# Postoperative Epileptic Seizures in Children

**DOI:** 10.3390/children9101465

**Published:** 2022-09-24

**Authors:** Luca Massimi, Paolo Frassanito, Federico Bianchi, Luigi Fiorillo, Domenica Immacolata Battaglia, Gianpiero Tamburrini

**Affiliations:** 1Neuroscience Department, Catholic University Medical School, 00168 Rome, Italy; 2Pediatric Neurosurgery, Fondazione Policlinico Universitario A. Gemelli IRCCS, 00168 Rome, Italy; 3Pediatric Neurology, Fondazione Policlinico Universitario A. Gemelli IRCCS, 00168 Rome, Italy

**Keywords:** epilepsy, postoperative seizures, brain tumors, neurosurgery, supratentorial lesions, pediatric patients

## Abstract

Background: Postoperative seizures (PS) occur in 10–15% of patients. This study aims to provide an update on the role of surgery in PS. Methods: All children undergoing a craniotomy for supratentorial lesions in the last 10 years were considered except those with preoperative seizures, perioperative antiepileptic drugs prophylaxis, head-injury and infections, repeated surgery, or preoperative hyponatremia. Children undergoing surgery for intra-axial lesions (Group 1, 74 cases) were compared with those harboring extra-axial lesions (Group 2, 91 cases). Results: PS occurred in 9% of 165 cases and epilepsy in 3% of 165 cases (mean follow-up: 5.7 years). There was no difference between the two study groups with regard to demographic data or tumor size. Group 1 showed a higher rate of gross total tumor resection (*p* = 0.002), while Group 2 had a higher rate of postoperative hyponatremia (*p* < 0.0001). There were no differences between the two groups in the occurrence of seizures (6.7% vs. 11%) or epilepsy (2.7% vs. 3.2%). No correlations were found between seizures and age, tumor location, histotype, tumor size, or the extent of tumor resection. Hyponatremia affected the risk of PS in Group 2 (*p* = 0.02). Conclusions: This study shows a lower rate of PS and epilepsy than series including children with preoperative seizures. Hyponatremia has a significant role. Neurosurgery is safe but surgical complications may cause late epilepsy.

## 1. Introduction

The occurrence of epileptic seizures after an intracranial neurosurgical operation is a well-known phenomenon, occurring in 5% to 20% of patients undergoing brain surgery [1,2,3,4,5,6,7,8]. Several risk factors have been propounded to explain such an event [9,10,11,12,13,14,15]: (1) the type of lesion (low grade gliomas, meningioma, abscess, aneurysm, or chronic subdural hemorrhage); (2) its location; (3) an age less than 2 years; (4) some comorbidities (e.g., postoperative electrolytic imbalance in children, presence of cognitive impairment in adults); (5) cortical damage related to the surgical approach. Usually, data about postoperative seizures are deduced from analyses based on surgical series which are presented for different reasons. Instead, studies specifically focusing on this problem are rare in children and often they include patients with preoperative seizures [16].

The present study aims to investigate whether the advent of better surgical techniques and more appropriate postoperative management have reduced the incidence of this complication and the possible role of the surgical operation in its pathogenesis. The goal is also to provide some more information about the role of other risk factors and on the still debated role of the perioperative antiepileptic drug prophylaxis. This study exclusively takes into account children and patients without preoperative seizures.

## 2. Materials and Methods

All children consecutively admitted to our institution between January 2010 and December 2019 who were undergoing surgery for supratentorial lesions were enrolled for this study and retrospectively reviewed. This time period was considered to have a homogeneous preoperative and postoperative work-up and management, as well as a minimum follow-up of 2 years. Patients with preoperative seizures were excluded as well as those receiving perioperative anti-seizure medications (ASMs), prophylaxis, harboring acute lesions predisposing them to seizures (head-injury, brain abscess and other infectious related lesions, preoperative evidence of electrolytic imbalance, early recurrence of disease), and those undergoing repeated surgery. Finally, children undergoing noncraniotomic neurosurgical procedures (e.g., neurendoscopy, transnasal endoscopy) or shunting procedures or with other prosthesis left in place, were excluded as well as those lost at follow-up. All patients received preoperative routine blood examinations, including sodium levels screening.

The recruited patients were divided into two groups to assess the role of the brain “violation” (cortical incision) compared with more anatomical routes (brain retraction). Accordingly, Group 1 was composed of children who underwent a corticotomy/corticectomy for subcortical or deep lesions (e.g., transcortical route for high-grade glioma), while Group 2 included those who underwent an intracranial approach for “extra-cerebral” lesions not requiring a cortical incision (e.g., trans-Sylvian route for optic pathways glioma). The cortico–subcortical or deep location (e.g., thalamic or intraventricular region) of lesions in Group 1 required a transcortical approach, with a brain incision or corticectomy and mild brain retraction. Instead, the mainly cisternal location of lesions of Group 2 required different approaches that were more “anatomical” but, at the same time, required a more significant brain retraction (children are characterized by a “full” brain). The substantial difference between the two types of approach is the main reason for the aforementioned grouping choice.

PS were categorized as immediate (occurring within 24 h of surgery), early (within one week), and late (after one week). Seizures were classified according to ILAE 2017 classification [17]. Children presenting postoperative seizures underwent an EEG and a specific neurological follow-up.

The following variables were considered in the statistical analysis (χ^2^ test and 2-tailed *t*-test) which aimed to assess possible differences between the two groups and the possible correlation between PS and risk factors inside each group: patients’ ages (namely, < and >2 years) and sex, location of the lesion, size of the lesion, histotype, extent of the tumor/lesion resection, and hyponatremia/electrolytic imbalance. Any *p* value < 0.05 was considered as statistically significant.

## 3. Results

### 3.1. Characteristics of the Two Groups

In the considered period, about 4500 children were operated on. One fifth of them (917 cases) underwent surgery for brain lesions and, finally, 165 were eligible for the present study (18%). Overall, 752 children were excluded for the following reasons: preoperative seizures (141 cases), therapy with perioperative anti-seizure medications (82 cases), acute lesions predisposing them to seizures (169 cases), repeated surgery (81 cases), operation by transnasal endoscopy (74 cases), incomplete follow-up (205 cases). All patients received a craniotomic approach. Their main findings are summarized in Table 1.

74 patients (38 boys and 36 girls) composed Group 1, with a mean age at surgery of 11.2 years (range: 2 weeks–18 years). The brain hemispheres and the ventricles were the most commonly involved regions because astrocytomas, choroid plexus tumors, and ependymomas were the most frequent histotypes. The number of lesions with diameters less or more than 3 cm was roughly equivalent. A gross total/near total removal of the tumor was possible in 80% of the cases. In all instances, a cortical incision was realized to get and remove the lesion (corticotomy or corticectomy). No significant intraoperative complications occurred. Postoperative electrolytic imbalance with transient hyponatremia affected 8% of cases. Other postoperative complications were represented by: CSF leakage (2 cases, 2.7%), subdural hygroma (2 cases, 2.7%), cortical infarction (2 cases, 2.7%), wound infection (1 case, 1.3%), and transient behavior disturbance (1 case, 1.3%). All but 7 children (90.5%) were alive at most current follow-up (mortality burdened mainly patients with high-grade gliomas, ependymomas, and AT/RT). Permanent neurological morbidity affects 4 children (5.5%), one of them with postoperative epilepsy (see below).

91 patients belonged to Group 2 (50 boys and 41 girls). Their mean age at surgery was 10.6 years (range: 1 month–18 years). Craniopharyngioma was the most common histotype followed by optic pathway gliomas, pineal tumors, and arachnoid cysts. Therefore, the sellar/suprasellar region was widely the most frequently involved area (66%). Given the complexity of this region, the larger number of lesions with a diameter more than 3 cm (63%), and that some types of tumor are hard to completely resect (namely, optic pathway gliomas), gross total/near total resection of the tumor was obtained in 57% (arachnoid cysts are not counted because they were treated by microsurgical fenestration alone). For similar reasons (namely, a high number of tumors involving the hypothalamus), the rate of postoperative hyponatremia or sodium imbalance was as high as 43%. A relevant intraoperative complication occurred in a child with craniopharyngioma (rupture of the right A2-segment of the anterior cerebral artery), without permanent sequelae nor epilepsy. CSF leakage occurred in 5 cases (5.5%), CSF infection in 2 cases (2.1%), cortical infarction and subdural hygroma in 3 cases each (3.2%), transient behavioral problems in 3 cases (3.2%), worsening of visual functions in 4 cases (4.4%), worsening of endocrinological imbalance in 7 cases (7.7%), and obesity in 8 cases (8.8%). All but 6 children (93.5%) were alive at most current follow-up (mortality burdened mainly patients with optic pathways gliomas, craniopharyngioma, and chordoma). Permanent neurological morbidity affects 14 children (15.3%), one of them with postoperative epilepsy (see below).

### 3.2. Postoperative Seizures and Epilepsy

Overall, PS occurred in 9% of cases (15 children). Focal seizures, with or without secondary generalization, accounted for 53% of all PS (8/15 cases), while generalized seizures covered the remaining 47%. The details are summarized in Table 2. Early seizures (7/15, 46.8%) were more frequent than immediate and late seizures (4/15, 26.6% each). Epilepsy burdened 5 children (3% of the whole series), two with focal to generalized tonic–clonic seizures, two with primary generalized tonic–clonic seizures and one with absence seizures. The diagnosis of epilepsy was made at a mean of 3.4 months after surgery. No epileptic syndromes occurred. EEG patterns varied according to the patients’ characteristics and the brain lobe harboring the epileptic foci, mainly with intercritical slowing activity, interictal spikes, or sharp waves (elicited by sleep), as well as irregular slow activity and epileptiform activity during seizures. The follow-up ranges from 2.5 to 10.5 years (mean: 5.7 years). 

Postoperative seizures were detected in 6.7% of cases in Group 1 (5/74 children) and 11% of cases in Group 2 (10/91 children), while epilepsy occurred in 2.7% and 3.2% of cases, respectively.

More specifically, Group 1 patients showed immediate and early seizures in two cases each, and late seizures in the remaining case (Table 2). They were mainly focal (3/5 cases) and multiple seizures (3/5 cases), with single seizures occurring in two cases (no patient had hyponatremia). Transient ASMs were used in case of multiple seizures. Only two patients ultimately developed epilepsy requiring prolonged ASMs (none with postoperative hyponatremia). One of them (focal to generalized tonic-clonic seizure) had the postoperative course complicated by subdural hygroma treated conservatively, while the remaining one (generalized tonic–clonic seizures) developed a late cortico–subcortical glio-malacic degeneration at the level of the surgical field (Figure 1). After a 5.7-year mean follow-up, both children still require ASMs: the first one is seizure-free, while the second one underwent vagal nerve stimulation for refractory epilepsy. The remaining patients are seizure-free and drug-free.

In Group 2, early seizures (5/10 cases) were prevalent with late (3/10) and immediate seizures (2/10) being less prevalent (Table 2). Purely focal seizures occurred in 2 out of 10 cases, the remaining patients showed primary or secondary generalized seizures (7 children had hyponatremia). Mainly, multiple seizures were evident (6 cases, 5 of them requiring transient ASMs). Three children (3.2%) ultimately developed epilepsy (focal to generalized tonic–clonic, generalized tonic–clonic, and absence in one each), at a mean of 3.5 months after surgery. Two of them (one with optic–hypothalamic astrocytoma and one with craniopharyngioma) had the postoperative course complicated by electrolytic imbalance with hyponatremia due to SIADH. The remaining patient (a young boy with tentorial atypical meningioma) developed postoperative occipital subcortical glio-malacic degeneration. At late follow-up, all of them are seizure-free; two of them still receive ASMs, while the treatment has been discontinued in the boy with craniopharyngioma (Figure 2). The remaining patients are seizure-free and drug-free.

### 3.3. Risk of Postoperative Seizures

There were no statistical differences between Group 1 and Group 2 as far as demographic data and tumor size were concerned. A significant difference was detected in: the extent of tumor resection, the gross total/near total resection being possible in 80% of cases in Group 1 versus 57% of Group 2 (*p* = 0.002); and hyponatremia, which occurred in 9% of cases in Group 1 versus 43% of cases in Group 2 (*p* < 0.00001) (Table 1).

No differences between the two groups were present in the frequency of PS and postoperative epilepsy (Table 2).

In each group, no correlation was found between the occurrence of seizures/epilepsy and: (1) age (infants <2 years versus children >2 years); (2) location (hemispheric vs. other, suprasellar vs. other); (3) histotype (gliomas versus other tumors in Group 1, craniopharyngioma versus other tumors in Group 2, high grade versus low grade tumor in both groups); (4) size of the tumor (≤3 cm versus >3 cm); (5) extent of tumor resection (gross total versus subtotal/partial). A significant correlation was found in Group 2 between hyponatremia and PS (*p* = 0.02). No correlation between other surgical complications and seizures was found.

## 4. Discussion

### 4.1. Epidemiology

The real frequency of PS is hard to estimate. Especially when immediate and/or isolated, they may be not adequately recorded and, subsequently, underreported. Moreover, the variability of the treated lesions and the adopted surgical techniques prevent a reliable assessment and comparison of the various series. Furthermore, most of the reported series include patients who already showed preoperative seizures and received ASM prophylaxis [16,18,19]. Taking all these limitations into account, the incidence of PS described in the literature ranges from 5% to 20% of cases in adults and 5% to 15% in children, with an estimated frequency of postoperative epilepsy of about 5–7% [1,2,4,6,11,20,21,22,23]. These figures quite often are results from studies carried out in the past, the results of which are still cited in recent articles. As mentioned, specific studies in children are missing, with most of the current papers being focused on adults (often on the risks related to surgery for meningiomas) [24,25,26] or on the use of ASM prophylaxis [27,28,29]. The few studies on children usually comprise mixed series including either children with or without preoperative seizures. For example, the exhaustive analysis provided by Saadeh and colleagues on 200 pediatric patients undergoing surgery for supratentorial tumors revealed a 34% rate of PS but only 16.5% of children had strictly PS [16].

The main goal of the present study was to assess the risk of seizures following surgery. For this reason, the spectrum of exclusion criteria for enrolling patients was wide, since not only patients with preoperative seizures or receiving ASMs were excluded but also all those where a perioperative or postoperative disease (head-injury, infection, hydrocephalus, early recurrence of the tumor, etc.) or the use of intracerebral prosthesis could affect this evaluation. The exclusion of patients with preoperative seizures is particularly important for the reliability of our study because a relevant proportion of all pediatric brain tumors (12%) and up to 75% of hemispheric brain tumors are diagnosed after the occurrence of preoperative seizures [30,31]. As a result, the present paper provides an update of the aforementioned figures, showing an overall incidence of postoperative seizures and epilepsy of 9% and 3%, respectively. These figures are, on average, lower than those reported in the literature since, as mentioned, they result more strictly from the effects of surgery than from other predisposing factors. The current figures are even lower that those presented in a previous personal study that was realized to assess the role of the cortical incisions in the genesis of postoperative seizures [32]. That analysis revealed a 13.2% overall rate of postoperative seizures and a 5.6% overall rate of epilepsy, without differences between patients undergoing a cortical incision and those who did not. Although, a reliable comparison between the two series is not possible. Since different populations of patients and diagnostic and surgical techniques are involved, the decrease in the incidence of seizures in the new series could reflect an improvement in the management of neurosurgical children due to technical and technological advances. A further explanation could be found in the more homogeneous materials and methods; in the more numerous sample size of patients; and in the more condensed analysis period that characterizes the present study.

### 4.2. Onset and Type of Seizures

After the definition given by Jennet more than 40 years ago about post-traumatic epilepsy, PS can be classified in terms of immediate, early, and late seizures [33]. Although immediate and early seizures are common among neurosurgical patients [1,34], late seizures are considered more prone to evolve into epilepsy as result of an iatrogenic epileptic focus formation [13]. The present series confirms findings in the literature regarding onset and type of seizures. Persisting epilepsy occurred at a mean of 3.4 months from the operation (no patient developed epilepsy later than one year after surgery) and its semiology was prevalently that of focal seizures. There were no significant differences between Group 1 and 2. Actually, according to the few studies available on this subject, the great majority of operated-on patients experience postoperative epilepsy within the first year after surgery (usually within 3 months) and only occasionally later [2,13]. Such an early onset has been explained by taking into account possible pathogenic mechanisms, hyponatremia, brain manipulation, or immediate surgical complications which would account for early postoperative epileptogenic damage. The prevalence of immediate/early seizures (73.4%) over late seizures (26.6%) in our study, compared with similar series where this relationship is inverse [16,35], can be explained once again by the adopted exclusion criteria, favoring the detection of purely “surgery-induced” seizures.

The aforementioned mechanisms could also explain the type of seizures, usually resulting from localized brain damage (focal seizures, focal to generalized seizures) or sodium imbalance (both focal and generalized seizures). In our experience, as well as in that of other authors [11,15,16], focal seizures were slightly predominant, namely in the short-term postoperative period (see also Table 2). The prevalence of focal seizures in Group 1 could be justified by the focal approach through the brain as well as the higher number of (generalized) seizures in Group 2 could depend on the more extensive and complex operations (also leading to electrolytic imbalance). On the other hand, one could speculate that the prevalently generalized pattern of the late epilepsy is mainly due to more extensive or multiple epileptic foci resulting from surgical complications.

### 4.3. Risk Factors

The etiology of PS represents a second problem that has been investigated quite extensively in the literature concerning the adult population. Accordingly, histotypes as glioma (brain infiltration) and meningioma (brain compression), vascular (AVM and aneurysm), and infectious lesions (brain abscess) are considered important risk factors in adults, especially when a preoperative cognitive impairment is associated [6,7,8]. Some specific studies on large series also pointed out the presence of consistent residual tumors (partial resection/biopsy) and the use of intraoperative cortical stimulation as two important risk factors [22]. As expected, such a risk must be evaluated taking into account the occurrence of multiple, systemic diseases which can contribute to the genesis of PS in adults [8,13]. Actually, even patients undergoing evacuation of subdural hematoma present a significant rate of postoperative seizures (14–18%) as result of the comorbidities of the elderly [9,10,36].

In children, the correlation between the type of lesion and postoperative seizures is less clear. Hardesty and colleagues, in a series of 223 operated-on children, found an incidence of perioperative epilepsy of 7.4% without any specific relationships with histotype, length of the surgical procedure, and amount of associated blood losses. Instead, the authors found supratentorial location, age < 2 years, and hyponatremia to be the only independent factors associated with perioperative seizures [11]. Conversely, the other study available in the literature on this topic, provided by Saadeh et al., showed a significantly increased risk of PS in children with high-grade tumors (versus low-grade) and with embryonal and pineal tumors (versus suprasellar tumors) in univariate analysis [16]. The same authors, in multivariate analysis, confirmed the role of an age < 2 years as seizure risk factor, together with preoperative seizures, hydrocephalus, and temporal lobe location. This study also reinforced the evidence of an important role of hyponatremia and a trivial role of the extent of tumor resection in favoring PS.

Our experience supports part of these observations. The first, most relevant matching point concerns hyponatremia that, in our study, was the only complication affecting the seizure/epileptic outcome. We did not include patients with preoperative electrolyte imbalance because, in this instances, hyponatremia is not a consequence of surgery, while the sodium imbalance following the hypothalamic manipulation is an effect of surgery. This factor remains a challenging management problem in the pediatric age. Moreover, in a specific study, it has been observed how hyponatremia affected postoperatively 12% of 319 children undergoing a neurosurgical operation for brain tumors, causing seizures and mental status alteration in 21% and 41% of cases, respectively [37]. In our series, hyponatremia was associated with late postoperative epilepsy in 2 out of 5 cases and was the main cause of transient early seizures in Group 2 (where the hypothalamus–pituitary axis was often involved by the tumor or manipulated during surgery). A second, matching point with previous studies is the missing role of the extent of tumor removal. Differently from adults, the extent of tumor resection does not seem to affect the occurrence of seizures in children, neither from a surgical (the more extensive approach necessary for a gross total resection does not increase the risk of seizures) nor from an oncological point of view (a larger amount of residual tumor, resulting from a partial resection, does not increase the risk of seizures). In the present study, no significant differences between the two Groups were found, in spite of the significant lower rate of gross total tumor resection obtained in Group 2. Similarly, the diameter of the lesion did not affect the prognosis, as observed in our previous study [32]. The tumor size, once again, seems to play a role mainly in adults undergoing surgery for meningiomas [18]. A further but partial matching point is represented by the histotype, which was not relevant in our study as was the case in Hardesty et al. [11] and for other authors [5] but was significant in the Saadeh et al. analysis [16]. This could represent a limitation of the present study where the strict criteria of patient selection (all patients presenting with preoperative seizures, which are one of the most common signs in those harboring brain tumors, were excluded) made the samples homogeneous for the postsurgical evaluation but less representative as far as the histotypes are concerned.

Finally, the main difference between the previous two studies and the present one concerns ages < 2 years, which was relevant for the aforementioned authors [11,16] but it was not in our series as well as in other studies [15,23]. Such a difference probably results from the small number of infants and the absence of patients with preoperative seizures characterizing our series. Infants with brain tumors remain a population predisposed to PS because of the quickly occurring brain changes and the prevalence of the influence of the excitatory network on the inhibitory one [38].

The tumor location deserves a final mention. As for other authors, we did not find differences related to the tumor location as far as the different regions of the supratentorial area are concerned [5,32]. Compared with other studies where, as expected, the temporal lobe plays a significant role [16,39], the present analysis did not confirm this data probably because of the relatively small number of patients with involvement of the temporal lobe. It is interesting to note that children operated on for posterior fossa tumors also can show PS, though less frequently (1.8–6% in spite of ASMs prophylaxis), as result of metabolic problems (acidosis, hyponatremia) or air embolism due to the sitting position [23,40].

### 4.4. Pathogenesis

PS are supposed to result from an acute cortical damage. More than one fifth of subjects with a head injury causing cortical damage, for example, present spontaneous seizures within 2 years from the trauma [41]. The intrinsic mechanisms of the epileptogenesis would be mainly mediated by the oxidative stress/free radicals formation (caused by extravascular leakage of blood components) and membrane ion imbalance (triggered by hypoxic–ischemic injury) [13,42,43,44]. Similar mechanisms can be hypothesized in case of the cortical injury resulting from the neurosurgical operation, where the microhemorrhages due to the cortical incision and the local ischemic damage resulting from the coagulation and/or the brain retraction (edema) can induce the aforementioned alterations, even though these modifications do not necessarily cause postoperative seizures [26]. The surgical incision of the brain can be considered per se as an epileptogenic stimulus. In a previous study that, to date, is the only one focusing on this specific topic in the literature, we did not find differences between children operated on through a cortical incision and those approached through a more anatomical route (like the Sylvian fissure or the suboccipital transtentorial route) [32]. In the present study, the same division into two groups (Group1/brain incision versus Group 2/brain retraction) was maintained. The current results confirmed that corticotomy/corticectomy does not increase the risk of PS, even in case of postoperative cortical infarction. Indeed, Group 1 children showed even a lower rate of both PS (6.7% vs. 11%) and epilepsy (2.7% vs. 3.2%) compared with Group 2, though this data did not reach a statistical significance. The study suggests that manipulation of certain cerebral regions associated to brain retraction can be more dangerous than cortical incision as far as postoperative seizures are concerned.

These findings suggest that the low epileptogenicity of the transcortical approaches could result from the advances in the neurosurgical technique and technology and from the increasing attention paid by the neurosurgeons to the neuroprotection during the operation. The improvement in terms of PS and epilepsy, which was noticed by comparing the current experience (PS: 9%, epilepsy:3%) with the previous experience that ended about 10 years ago (PS: 13.2%, epilepsy: 5.6%), supports this hypothesis. On the other hand, brain distortion, which can occur during the noncorticotomic approaches, may result in more danger than a cortical incision. Actually, a brain retraction, possibly causing postoperative edema, infarction, and cortical atrophy, is often performed in extracerebral approaches in children, whose brain is trophic but frail and the size of arachnoid cisterns is particularly small. The pathogenetic role of brain manipulation or retraction is indirectly demonstrated by the lower incidence (1%) of postoperative seizures following endoscopic transsphenoidal surgery as compared with craniotomic approaches [45]. In some experimental animal models, the craniotomy alone was enough to increase the risk of PS because of the induced edema [46].

### 4.5. Perioperative ASMs Prophylaxis

The present study provides a little contribution to the undergoing discussion concerning indications for ASM prophylaxis [7,12,14,28,41,47,48]. Actually, the rate of children with PS after corticotomy or noncorticotomic approaches in the series considered here, in which no antiepileptic prophylaxis was adopted, is similar or even lower to that observed in surgical series where ASM prophylaxis was administered [23,49]. Most of the current studies and analyses from the literature do not support the use of preoperative or perioperative ASM prophylaxis due to the lack of evidence of clinical advantages [27,50,51]. At most, this prophylaxis could be considered in the cases of high-risk patients and according to the context of each institution (e.g., use of intraoperative cortical stimulation) [22]. The updated Cochrane review on this topic (2020) failed to show if ASM prophylaxis is effective or not in preventing postsurgical early or late seizures, to due to the limited, low-certainty evidence [28]. Therefore, good-quality, contemporary trials are needed to provide closure to this debate.

## 5. Conclusions

The present study shows a lower rate of seizures and de novo onset epilepsy than in the past following pediatric intracranial neurosurgical operations. This rate is lower than that observed in adults undergoing similar procedures or after head injury. Among the possible risk factors, hyponatremia plays a significant role. Since this study was mainly focused on the role of surgery, other factors should also be considered according to the literature, such as an age < 2 years, preoperative seizures, and temporal lobe location. The surgical operation does not increase the risk of PS, but neurosurgical complications may lead to postoperative epilepsy.

## Figures and Tables

**Figure 1 children-09-01465-f001:**
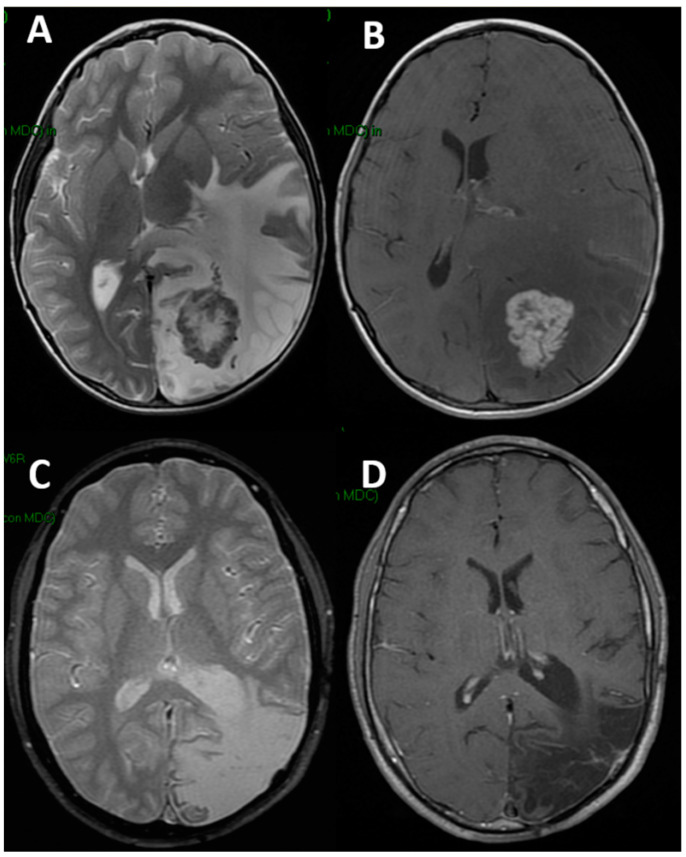
Preoperative brain MRI ((**A**): axial T2-weighted; (**B**): axial T1-weighted after gadolinium) of a 9-year-old boy harboring an intra-axial schwannoma. Note the extensive perilesional edema and the mass effect. The same sequences (**C**,**D**) five years later: no tumor recurrence but glio-malacic cortical involution is present. The patient is relying on ASMs and vagal nerve stimulation.

**Figure 2 children-09-01465-f002:**
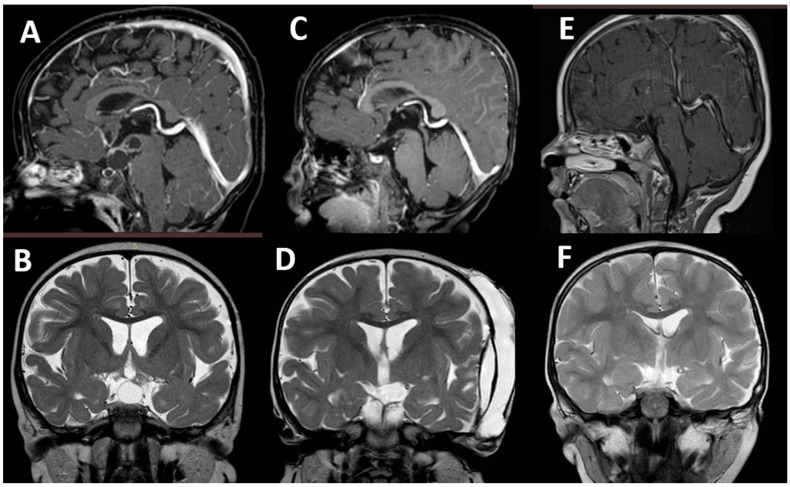
Preoperative MRI ((**A**): sagittal T1 after gadolinium, (**B**): coronal T2) of a 6-year old boy with craniopharyngioma with sellar/suprasellar extension and extended up to the interpeduncular cistern. The immediate postoperative MRI ((**C**,**D**), same MRI sequences) shows the tumor was removed without significant brain damage but with subdural hygroma and subcutaneous CSF collection. The patient developed postoperative epilepsy associated to hyponatremia. ASMs were discontinued 4 years after surgery. At that time, MRI was normalized ((**E**,**F**), same MRI views).

**Table 1 children-09-01465-t001:** Main findings in the two groups.

	*Group 1*	*Group 2*
*No*	74	91
*M/F ratio*	1.05	1.21
*Mean age at surgery*	11.2 years	10.6 years
*Children ≤ 2 year-old)*	10 (13.5%)	11 (12%)
*Type of lesion*	High grade gliomas: 10 (13.5%)	Craniopharyngioma: 38 (42%)
	Low-grade gliomas: 17 (23%)	Pituitary adenoma: 2 (2%)
	DNET: 7 (9.5%)	Optic glioma: 16 (17%)
	Choroid plexus tumors: 14 (19%)	Pineal tumors: 14 (15%)
	Ependymomas: 9 (12%)	Arachnoid cyst: 10 (11%)
	Cavernous angioma: 8 (11%)	Chordoma: 4 (5%)
	AT/RT: 5 (6.5%)	Dermoid cyst: 4 (5%)
	Other: 4 (5.5%)	Other: 3 (3%)
*Region*	Hemispheric: 36 (48.5%) Frontal: 18 Temporal: 6 Parieto-occipital: 12 Intraventricular: 27 (36.5%) Thalamic: 11 (15%)	Sellar/suprasellar: 60 (66%) Pineal: 14 (15%) Middle fossa: 11 (12%) Interhemispheric: 3 (3.5%) Retro-orbital: 3 (3.5%)
*Max diameter*		
*≤3 cm*	39 (52%)	34 (37%)
*>3 cm*	35 (48%)	57 (63%)
*Extent of tumor removal*		
*Gross/near total*	59 (80%)	46 (57%) *
*Subtotal/partial*	15 (20%)	35 (43%) *
*Postop hyponatremia*	6 (8%)	39 (43%)

* Not including arachnoid cysts (which underwent only microsurgical fenestration).

**Table 2 children-09-01465-t002:** Postoperative seizures.

	*Group 1*	*Group 2*	*Total*
No.	74	91	165
Postop seizures	5 (6.7%)	10 (11%)	15 (9%)
Timing and type			
Immediate	2: FGTC + GT	2: both FGTC	4: FGTC (3) + GT
Early	2: FM + GTC	5: FM + FNM + GT + GTC (2)	7: FM (2) + FNM + GT + GTC (3)
Late	1: FM	3: A + FGTC + GTC	4: A + FM + FGTC + GTC
Postop epilepsy	2 (2.7%)	3 (3.2%)	5 (3%)
Type and time from surgery	FGTC: 2.5 month GTC: 4 months	A: 5 months FGTC: 2 month GTC: 3.5 months	A + FGTC (2) + GTC (2) Mean time from surgery: 2.5 months

A: absence (generalized), FM: focal motor seizure, FNM: focal non-motor seizure, FGTC: focal to generalized tonic–clonic seizure, GTC: generalized tonic-clonic seizure, GT: generalized tonic seizure.

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
