# Peer review of "Postoperative Epileptic Seizures in Children"

_children, 2022, doi:10.3390/children9101465_

Round 1

Reviewer 1 Report

The objective is to know the risk of postoperative seizures and epilepsy, considering two groups of patients: those who underwent a cortical incision (Group 1)  and those in which the brain was retracted but not 'violated' (Group 2). A rigid selection control was observed and, patients with a predisposition to or preoperative seizures were excluded as well as those using anti-epileptic drugs or other with preexistent brain diseases. The conclusion was that  hyponatremia affected the risk of seizures in Group 2

The article as a whole is very well prepared and it is important to point out that this subject has been little explored in dealing with epilepsy in childhood.

A few suggestions to the authors:

Abstract: In Results (line 15 and following), it would become more explicit and emphatic if the sentences starting with "No differences...; no correlation..." was preceded by an appropriate verb form, in the case "There was ..."

Results: It would be interesting to make it clear if there was a preoperative screening of blood sodium levels for all patients in both groups or whether only those who developed postoperative seizures were tested. This did not seem clear to me in the text.

Tables 1 and 2. Unless the Tables are according to the standards of the Journal, I suggest organizing them with columns starting at the left and not at the center.

Congratulations to the authors for their work.

Author Response

The objective is to know the risk of postoperative seizures and epilepsy, considering two groups of patients: those who underwent a cortical incision (Group 1)  and those in which the brain was retracted but not 'violated' (Group 2). A rigid selection control was observed and, patients with a predisposition to or preoperative seizures were excluded as well as those using anti-epileptic drugs or other with preexistent brain diseases. The conclusion was that  hyponatremia affected the risk of seizures in Group 2

The article as a whole is very well prepared and it is important to point out that this subject has been little explored in dealing with epilepsy in childhood.

  • Answer: thank you for the comment and the review.

A few suggestions to the authors:

Abstract: In Results (line 15 and following), it would become more explicit and emphatic if the sentences starting with "No differences...; no correlation..." was preceded by an appropriate verb form, in the case "There was ..."

  • Answer: we modified the text accordingly.

Results: It would be interesting to make it clear if there was a preoperative screening of blood sodium levels for all patients in both groups or whether only those who developed postoperative seizures were tested. This did not seem clear to me in the text.

  • Answer: both groups received preoperative sodium screening as a routine evaluation. This information has been added in the manuscript.

Tables 1 and 2. Unless the Tables are according to the standards of the Journal, I suggest organizing them with columns starting at the left and not at the center.

  • Answer: not sure to have got the question. The columns start at the left

Congratulations to the authors for their work.

  • Answer: thank you again!

All changes are reported in red in the manuscript.

Reviewer 2 Report

The information the readers can appreciate from this study is only the incidence of seizures after intracranial surgery.

Description of presence of intra- or postoperative complications, EEG findings for patients who developed seizures, type of epilepsy in patients who developed epilepsy, and clinical courses after surgery for all patients, and other factors that can contribute the development of seizures should be considered to be presented to improve the study.

Author Response

The information the readers can appreciate from this study is only the incidence of seizures after intracranial surgery.

  • Answer: thanks for the comment and the review.

Description of presence of intra- or postoperative complications, EEG findings for patients who developed seizures, type of epilepsy in patients who developed epilepsy, and clinical courses after surgery for all patients, and other factors that can contribute the development of seizures should be considered to be presented to improve the study.

  • Answer: we included all the required information in the manuscript, unless already present (e. g. type of postoperative epilepsy)

All changes are in red in the new version of the manuscript

Reviewer 3 Report

The article aims to provide an update on the role of surgery in postoperative seizures, particularly for children. The study is interesting though some critical issues would need to be addressed.

1. In the abstract, the author stated that Children undergoing surgery for intra-axial lesions (Group 1, 74 cases) 13 were compared with those harboring extra-axial lesions (Group 2, 91 cases). PS occurred in 14 9% and epilepsy in 3% of 165 cases (mean follow-up: 5.7 years). I did not see relevant description in the main text. 

2. The authors say Hyponatremia affected the risk of PS, is it possible that the tumor location and the surgery affect the sodium metabolism?

3. In the introduction section at line 38-41, The present study aims at investigating whether the advent of better surgical techniques and more appropriate postoperative management have reduced the incidence of this complication and the possible role of the surgical operation in its pathogenesis. while no evidence is presented in the results section.

4. The recruited patients were divided into two groups to assess the role of the brain “violation” (cortical incision) compared with more anatomical routes (brain retraction). The authors should give more details in the grouping reason.

Author Response

The article aims to provide an update on the role of surgery in postoperative seizures, particularly for children. The study is interesting though some critical issues would need to be addressed.

- Answer: thank you for the comment and the review.

  1. In the abstract, the author stated that Children undergoing surgery for intra-axial lesions (Group 1, 74 cases) 13 were compared with those harboring extra-axial lesions (Group 2, 91 cases). PS occurred in 14 9% and epilepsy in 3% of 165 cases (mean follow-up: 5.7 years). I did not see relevant description in the main text. 

- Answer: more data about the two groups have now been included. Most of the information to compare the patients are inside the tables.

  1. The authors say Hyponatremia affected the risk of PS, is it possible that the tumor location and the surgery affect the sodium metabolism?

- Answer: sure! Actually, hyponatremia occurred mainly in group 2, where the hypothalamus was involved by the tumor or manipulated during surgery. This information was already discussed and it is now highlighted in the manuscript.

  1. In the introduction section at line 38-41, The present study aims at investigating whether the advent of better surgical techniques and more appropriate postoperative management have reduced the incidence of this complication and the possible role of the surgical operation in its pathogenesis. while no evidence is presented in the results section.

- Answer: the comment on that is present in the discussion.

  1. The recruited patients were divided into two groups to assess the role of the brain “violation” (cortical incision) compared with more anatomical routes (brain retraction). The authors should give more details in the grouping reason.

- Answer: new details have been added in the Methods section

All changes are reported in red in the manuscript.

Round 2

Reviewer 2 Report

Thanks for the modifications.

Author Response

Thank you

Reviewer 3 Report

The authors sufficiently addressed my concerns.

Author Response

Thank you